# Semi-Supervised Semantic Segmentation of Remote Sensing Images Based on Dual Cross-Entropy Consistency

**DOI:** 10.3390/e25040681

**Published:** 2023-04-19

**Authors:** Mengtian Cui, Kai Li, Yulan Li, Dany Kamuhanda, Claudio J. Tessone

**Affiliations:** 1College of Computer Science and Engineering, Southwest Minzu University, Chengdu 610041, China; 2Department of Science Mathematics and Physical Education, College of Education, University of Rwanda, Kigali P.O. Box 3900, Rwanda; 3Department of Informatics, University of Zurich, Andreasstrasse 15, CH-8050 Zurich, Switzerland

**Keywords:** cross-entropy consistency, information entropy, semi-supervised, channel attention mechanism, remote sensing image

## Abstract

Semantic segmentation is a growing topic in high-resolution remote sensing image processing. The information in remote sensing images is complex, and the effectiveness of most remote sensing image semantic segmentation methods depends on the number of labels; however, labeling images requires significant time and labor costs. To solve these problems, we propose a semi-supervised semantic segmentation method based on dual cross-entropy consistency and a teacher–student structure. First, we add a channel attention mechanism to the encoding network of the teacher model to reduce the predictive entropy of the pseudo label. Secondly, the two student networks share a common coding network to ensure consistent input information entropy, and a sharpening function is used to reduce the information entropy of unsupervised predictions for both student networks. Finally, we complete the alternate training of the models via two entropy-consistent tasks: (1) semi-supervising student prediction results via pseudo-labels generated from the teacher model, (2) cross-supervision between student models. Experimental results on publicly available datasets indicate that the suggested model can fully understand the hidden information in unlabeled images and reduce the information entropy in prediction, as well as reduce the number of required labeled images with guaranteed accuracy. This allows the new method to outperform the related semi-supervised semantic segmentation algorithm at half the proportion of labeled images.

## 1. Introduction

With the continued improvements in information technology, the sensor technology and space science technology involved in remote sensing imaging have also advanced, and remote sensing imaging technology now plays a critically important role in Earth observation. Remote sensing images provide information for a large number of observation tasks, and the advances in remote sensing image technology are driving the military [1,2], meteorological, and transportation fields. In recent years, the development of satellite imaging has been rapid, and remote sensing images have become more convenient to obtain; the image information has become more complex [3], and the remote sensing image data have grown dramatically. Thus, remote sensing images are now numerous and complex [4].

Convolutional neural networks simplify image processing tasks [5], and real-time intelligent image processing techniques provide the basis for the development of downstream tasks [6]. However, efficient deep learning models rely on supervised learning with large, manually labeled datasets [7]. A huge labeled dataset requires a lot of time, as well as large labor costs. The higher spatial resolution of remote sensing images [8], multiple categories, and complex image information lead to higher costs of labeling remote sensing datasets. The labels also used for the semantic segmentation task require pixel-level annotation, and the high annotation cost becomes one of the main problems limiting the development of semantic segmentation. Many scholars have started to explore the information contained in unlabeled images and explore the use of unlabeled images to train segmentation models, reducing prediction information entropy, which makes semi-supervised learning models a popular development in image segmentation.

Semi-supervised learning methods have achieved good results in the field of semantic segmentation in recent years [9], reducing the costs of labels needed to train models. Zhu et al. [10] trained a model with a few labeled images and then used the model to generate pseudo-labels of unlabeled images directly. Then, all data have a corresponding label or pseudo-label. The final dataset can then be used to train a new model, reducing the labeling costs. However, this method relies too much on the pseudo-label of the first model and the prediction results contain large information entropy. Tarvainen and Valpola [11] proposed an iterative training method, which used the average weights trained by the students as the new teacher model after each training step, and obtained good results after iterative training through several iterations. The shortcomings of the teacher model were corrected by the iterative method, but the iterative iteration introduced a large amount of computation.

The consistency regularization method proposed by Luo et al. [12] indicates that for the same pixel, after different perturbations, the information entropy in predictions should be consistent, and for the input after different disturbances, the information entropy in predictions should be consistent. This method places an entropy consistency constraint on the image predictions and is now a widely used method in semi-supervised learning. Ke et al. [13] processed the input images with different interference, went through two segmentation networks with different parameter initialization and an identical structure, and forced the information entropy in prediction consistency between the two networks. Zou Y et al. [14] proposed to classify the image perturbation into two types of strong and weak perturbation, and to use the prediction results of weak perturbation [15] processed as the pseudo-label of strong perturbation, because the prediction results after weak perturbation are more stable, and this novel method promotes the development of semi-supervised learning.

Chen X et al. [16] proposed a cross-pseudo supervision model (CPS) based on the above approach, and the predictions under different perturbation models are used as pseudo-labels for mutual supervision. This method not only has a clear model but also a good training effect, which fully exploits the hidden information in the unlabeled images. This method achieves significant results, but for the remote sensing images, the overlap rate between categories is high, and the local categories are many and complex; this also means that more comprehensive training is required. Wu et al. [17] designed a semi-supervised segmentation model consisting of an encoding [18] network and two different decoding networks based on the consistency regularization. The resultant bias of the two decoding networks is set to unsupervised loss, thus promoting prediction consistency between the two decoding networks and allowing the model to fully understand the large amount of information in unlabeled images.

In summary, the reasonable use of unlabeled images, reducing the information entropy of unsupervised predictions [19,20,21], enabling the model to fully exploit the hidden information in unlabeled images, and lowering the labeling costs are the keys of our research.

We propose the semi-supervised semantic segmentation of remote sensing images based on dual cross-entropy consistency with a model designed based on the teacher–student architecture. A channel attention (CA) mechanism [22,23] is added to the teacher model to filter the feature information and lower the information entropy of pseudo-label data. The student model with a dual decoding network through single coding networks ensures the consistency of the information entropy of the coding network results. The model is trained alternately through two tasks based on dual cross-entropy consistency, the pseudo-label of the teacher model, semi-supervision of the student models, and cross-supervision between the student models. This allows our method to exploit the hidden information in the unlabeled dataset, reduce the prediction information entropy, and lower the labeled image costs.

## 2. The Proposed Model

### 2.1. Semi-Supervised Segmentation Model Based on Dual Cross-Entropy Consistency

Our model includes a teacher model and dual student models. We use Unet [24] as our basic convolutional neural network because of its symmetric structure. The teacher model adds a CA mechanism to the coding network to filter feature information, highlight target features, and suppress noisy information, thus reducing the information entropy of unsupervised prediction. The two student models share a common coding network; the dual-decoding network architecture ensures that the output vectors of the coding network have consistent information entropy. A sharpen function [25] is used to reduce the information entropy of the unlabeled images’ predictions, and to improve the confidence of edge contours. The model is shown in Figure 1 below.

In each round of training, the dataset is divided according to the labeled set. We first train the teacher model with the labeled set, and the supervised loss is calculated by the ground truth and the parameters are updated. Next, the unlabeled set is used to generate pseudo-labels [26] by using the Hadamard product [27] and linear transformations of the predicted results and original images. We use these pseudo-labels to semi-supervise the predictions of the student models S1, S2, calculate the pseudo-supervised loss, and update the encoding network (S)–decoding network (S1) model and the encoding network (S)–decoding network (S2) model in turn. Finally, we obtains pseudo-labels (S1, S2) via the predictions of both student decoding networks and the original image, and update the parameters in turn via cross-supervision loss. The dual-entropy consistency tasks include a teacher model for the dual student models’ prediction information entropy consistency task and a cross-supervised entropy consistency task between two student models. The models are trained with alternating constraints by the two entropy consistency tasks, so that the model can fully understand the feature information in the unlabeled images and reduce the prediction information entropy.

### 2.2. Channel Attention Mechanism

In the task of the semantic segmentation of remote sensing images, the procedure often involves many categories and a complex topography, with many overlaps between categories, and the features between categories are not prominent, etc. The CA mechanism is widely used in remote sensing image processing because the CA mechanism can effectively filter the feature map and suppress noise interference. Therefore, we add the CA module to the teacher coding network to constrain the feature extraction and reduce the information entropy generated by the coding network. The CA mechanism is shown in Figure 2 below.

The mathematical description of the channel attention mechanism module is as follows. First, we perform adaptive global average pooling and adaptive global maximum pooling on the input feature map *F*($F=R^{H\times W\times C}$), respectively, and pass the results through the fully connected layer and the RELU function to obtain two vectors, $U_{avg}$ and $U_{\max}$, with global sense fields. The specific forms are shown in Equations (1) and (2).
(1)$$U_{avg}=FC_{2}\left(RELU\left(FC_{1}\left(avgPooling\left(F\right)\right)\right)\right),$$
(2)$$U_{\max}=FC_{2}\left(RELU\left(FC_{1}\left(\max Pooling\left(F\right)\right)\right)\right),$$

Subsequently, the two vectors are fused channel by channel and then activated by the sigmoid nonlinear function, as shown in Equation (Equation 3). This is because the maximum and average pooling can screen channels from different angles. After the fusing by the channel, the sigmoid nonlinear activation function can be used to obtain an ideal weight *U*. Finally, the weight *U* is multiplied channel by channel with the input feature map, as shown in Equation (Equation 4). The new feature map $F^{{}^{\prime }}$, generated after feature information screening, can be used for subsequent segmentation tasks by highlighting the effective feature information and suppressing invalid information.
(3)$$U=sigmoid(U_{avg}⊕U_{\max}),$$
(4)$$F^{{}^{\prime }}=U⊗F,$$

### 2.3. The Sharpen Function

In the semantic segmentation algorithm based on consistency regularization, scholars usually assume that the final prediction boundary should not pass through the high-density region of edge pixel distribution, requiring low-entropy output for unlabeled images. The method using pseudo-label supervision is a type of unsupervised learning, and the generated pseudo-label will have unclear details, a low confidence level, and high information entropy, so the sharpen function [11] for the prediction results of student models can maximize the entropy reduction. The sharpen function is shown in Equation (Equation 5).
(5)$$S\left(y,T\right)=\frac{{\left(y\right)}^{1/T}}{\sum_{i=1}^{K}{\left(y\right)}^{1/T}},T\in \left(0,1\right),$$
where *y* is the prediction result of the network, *K* is the number of channels of the network output, and *T* is a hyperparameter in the interval (0, 1).

### 2.4. Loss Function

Normally, we use the softmax operation to obtain the prediction; this is to ensure that the prediction is finally mapped to the (0, 1) interval. The most classical loss function for semantic segmentation, the pixel-level cross-entropy [28] loss, which is able to examine each pixel individually, compares the predictions for each pixel class with the label. The cross-entropy is defined by the following Equation (Equation 6).
(6)$$CE\left(y_{i},y_{i}^{{}^{\prime }}\right)=-\sum_{i}\left(y_{i}\log\left(y_{i}^{{}^{\prime }}\right)+\left(1-y_{i}\right)\log\left(1-y_{i}^{{}^{\prime }}\right)\right),$$
where $y_{i}^{{}^{\prime }}$ is the label class of any pixel *i*, and $y_{i}$ is the predicted result at *i*.

For the dataset *D*, we divide it into a labeled dataset $D^{l}$ of size *N* and an unlabeled dataset $D^{u}$ of size *M*. The initialized weights of the teacher encoding network ($E_{t})$ and the teacher decoding network ($D_{t}$) are $\theta_{e-t}$ and $\theta_{d-t}$. $D^{l}$ is used as supervised learning to train the teacher models. For input image *x*, the corresponding higher-order semantic vector $V_{x}$ is firstly obtained through the teacher model encoding network, and secondly the final prediction $P^{t}$ is obtained through the decoding network, as shown in Equations (7) and (8) below.
(7)$$V_{x}=E_{t}\left(x:\theta_{e-t}\right)\left(x\in D^{l}\right),$$
(8)$$P^{t}=D_{t}\left(V_{x}:\theta_{d-t}\right)\left(x\in D^{l}\right),$$

The supervised loss $L^{t}$ is calculated from the predicted results of the teacher model with labels, as shown in Equation (Equation 9).
(9)$$L^{t}=\frac{1}{N}\sum_{x\in D^{l}}\frac{1}{W\times H}\sum_{i=1}^{W\times H}l_{ce}\left(P_{i}^{t},Y_{i}^{*}\right),$$
where *W* and *H* represent the width and height of the input image, *i* represents any pixel of the output image, $P_{i}^{t}$ represents the predicted value of the prediction result $P^{t}$ at pixel *i*, $l_{ce}$ represents the loss function mentioned in Equation (Equation 6), $Y^{*}$ represents the ground truth of the input image *x*, and $Y_{i}^{*}$ represents the true class of pixel *i* in the label.

For unlabeled dataset $D^{u}$ of size *M*, the image *x* is first input to the teacher model to obtain the teacher prediction $P^{t}$, and the pseudo-label $Y^{t}$ is obtained by the input image *x* and $P^{t}$. For the student models, the initialized weights of the encoding network ($E_{s}$) is $\theta_{e-s}$ and the decoding network ($D_{s}$) are $\theta_{d-s1}$ and $\theta_{d-s2}$, respectively. For the input unlabeled image *x*, the corresponding higher-order semantic vector $V_{x}$ is firstly obtained by the student coding network, and the final prediction results are obtained by the two decoding networks. Reducing the output information entropy with the sharpen function, the final outputs $P^{s1}$, $P^{s2}$ are given by the following Equations (10)–(12).
(10)$$V_{x}=E_{s}\left(x:\theta_{e-s}\right)\left(x\in D^{u}\right),$$
(11)$$P^{s1}=S\left(D_{s}\left(V_{x}:\theta_{d-s1}\right)\right)\left(x\in D^{u}\right),$$
(12)$$P^{s2}=S\left(D_{s}\left(V_{x}:\theta_{d-s2}\right)\right)\left(x\in D^{u}\right),$$

The pseudo-supervised loss of $Y^{t}$ on $P^{s1}$, $P^{s2}$ was
(13)$$L^{t-s1}=\frac{1}{M}\sum_{x\in D^{u}}\frac{1}{W\times H}\sum_{i=1}^{W\times H}l_{ce}\left(P_{i}^{s1},Y_{i}^{t}\right),$$
(14)$$L^{t-s2}=\frac{1}{M}\sum_{x\in D^{u}}\frac{1}{W\times H}\sum_{i=1}^{W\times H}l_{ce}\left(P_{i}^{s2},Y_{i}^{t}\right),$$

In Equations (13) and (14), $L^{t-s1}$ and $L^{t-s2}$ represent the pseudo-supervised losses of the teacher model pseudo-label for the two student models, respectively; $P_{i}^{s1}$, $P_{i}^{s2}$ represent the predicted values of $P^{s1}$, $P^{s2}$ at pixel *i*. $Y_{i}^{t}$ represents the category of the pixel *i* in the label.

For the cross-supervision of the two student models, we create the pseudo-labels $Y^{s1}$, $Y^{s2}$ via the input image *x* and the final outputs $P^{s1}$, $P^{s2}$ of the two models. The prediction results corresponding to the cross-supervision of the two pseudo-labels are obtained as the cross-supervised loss.
(15)$$L^{s2-s1}=\frac{1}{M}\sum_{x\in D^{u}}\frac{1}{W\times H}\sum_{i=1}^{W\times H}l_{ce}\left(P_{i}^{s1},Y_{i}^{s2}\right),$$
(16)$$L^{s1-s2}=\frac{1}{M}\sum_{x\in D^{u}}\frac{1}{W\times H}\sum_{i=1}^{W\times H}l_{ce}\left(P_{i}^{s2},Y_{i}^{s1}\right),$$

$L^{s2-s1}$ and $L^{s2-s1}$ in Equations (15) and (16) above represent the supervisory loss of pseudo-label $Y^{s2}$ on student model S1 and the supervisory loss of pseudo-label $Y^{s1}$ on student model S2, respectively. $P_{i}^{s1}$, $P_{i}^{s2}$ represent the predicted values of $P^{s1}$, $P^{s2}$ at pixel *i*; $Y^{s1}$, $Y^{s2}$ represent the pseudo-labels of the two student models. $Y_{i}^{s1}$, $Y_{i}^{s2}$ represent the category of pixel point *i* in the pseudo-label in the label.

In summary, the semi-supervised loss includes the semi-supervised loss of the teacher to the dual students, and the cross-supervised loss of the dual students. We use this dual entropy consistency task to implement iterations of the student models so that the model fully understands the feature information in the unlabeled images, reduces the prediction information entropy, and improves the prediction accuracy.

## 3. Experiments

### 3.1. Experimental Dataset and Environment

In order to verify the effectiveness of the proposed semantic segmentation method for semi-supervised remote sensing images, we selected the Potsdam and Vaihingen datasets from the International Society for Photogrammetry and Remote Sensing (ISPRS) and the Gaofen 2 satellite image dataset (GID) from more than 60 cities in China. The Potsdam dataset contains 38 images in TIF format with a spatial resolution of 5 cm and a size of 6000 × 6000. The dataset is divided into six categories: impervious surface, building, low vegetation, tree, car, and clutter. The Vaihingen dataset contains 33 images in TIF format, with a spatial resolution of 9 cm. However, the image sizes are not consistent. The average size is 2494 × 2064 and the dataset is divided into the same six categories as Potsdam. The GID [29] dataset contains 150 images of the satellite with a size of 7200 × 6800. The size and format are consistent with the original images. The dataset is divided into five categories: farmland, forest, building, meadow, and water. Three datasets provide the corresponding labeled images for each image. For better experiments, all datasets are cropped to 512 × 512 size, and 10% of the dataset is selected as test images, while the rest of the images are used for model training.

Our experiments were implemented on a computer equipped with an NVIDIA RTX3060Ti GPU and INTEL 12400F CPU using the Pytorch framework. The batch size was set to 4, and the model was trained with the Adam optimizer with default parameters and aided by the Cosine warmup learning rate strategy [30]. The initial learning rate was set to 0.001, the number of training iterations was 100 epochs, and T was set to 0.5.

### 3.2. Evaluation Indicators

At present, academics usually measure the performance of semantic segmentation algorithms from three aspects: running time, memory occupation, and accuracy. Because accuracy is the most objective, we focus on the evaluation indicators of semantic segmentation accuracy. This mainly includes PA, MPA, Iou, MIou, recall, F1-score, etc. Among them, MIou is concise and representative, and it is the most commonly used indicator in the evaluation of semantic segmentation experiments. The definitions and calculation equations are detailed as follows.

(1) Iou: the ratio between the intersection of the predicted result and the ground truth. The definition is shown in Equation (Equation 17).
(17)$$Iou=\sum_{i=1}^{n}\frac{p_{ii}}{t_{i}+\sum_{j=1}^{k}\left(p_{ji}-p_{ii}\right)},$$

(2) MIou: the average value of the accumulated IoU values of each class of image pixels, as shown in Equation (Equation 18).
(18)$$MIou=\frac{1}{n}\sum_{i=1}^{n}\frac{p_{ii}}{t_{i}+\sum_{j=1}^{k}\left(p_{ji}-p_{ii}\right)},$$
where *n* represents the number of classes of pixels; $p_{ii}$ represents the number of pixels whose actual class is *i* and whose predicted class is *i*; $t_{i}$ represents the total number of pixels of class *i*; $p_{ji}$ represents the number of pixels whose actual class is *i* and predicted class is *j*.

### 3.3. Analysis of the Experimental Results

To verify the performance of our method, experiments were conducted on three datasets using different proportions of labeled images, and the method proposed was compared with the current popular semi-supervised and fully supervised methods. The comparison methods include three sets of fully supervised algorithms, Unet, Attention-Unet, and U2-Net [31]; and three sets of semi-supervised algorithms, Mean Teacher, CPS, and DST-CBC [32]. Table 1 gives the MIou performance for the related methods on three datasets.

Table 1 shows that our algorithm has poor training results for the teacher model when the label image proportion is low, the cross-training of the two student models cannot obtain good results, and the overall segmentation results are lower than other semi-supervised models. As the proportion of labeled images increases, alternate training of the dual-entropy consistency tasks shows an advantage, and when the proportion of labeled images reaches 1/2, the segmentation results of our algorithm surpass those of the other semi-supervised models. After introducing the sharpen function, the segmentation results of the model at the label image proportion of 1/2 are already higher than those of some fully supervised learning models. From the above results, we can see that our model can effectively improve the feature extraction efficiency of the model coding network and reduce the pseudo-label information entropy after adding the channel attention mechanism. The dual-entropy consistency tasks of the two student models are poor when the label image proportion is small, but as the proportion increases, the advantages of the dual-entropy consistency tasks are then reflected. Table 2 shows the MIou performance of our algorithm for each category on three datasets with different labeled image proportions.

The data in Table 2 shows that the MIou results for each class also conform to the overall distribution pattern, with a small performance improvement when the labeled image proportion is small. Our method has a larger rate of training result improvement as the labeled image proportion increases, which also saves a large part of the labeling cost. The effect of increasing the labeled image proportion is also found in the low vegetation class of the Vaihingen dataset and the forest class of the GID, which has a negative effect. This also means that we cannot simply increase the labeled image proportion and need to find the optimal connection between our dual cross-entropy consistency method and the labeled image proportion.

The prediction results on the three datasets are given in Figure 3, Figure 4 and Figure 5. The results show that the best segmentation is achieved on the dataset when the labeled image proportion is 1/2, with outstanding segmentation details, no obvious mis-segmentation and breakpoints, and minimum information entropy.

### 3.4. Ablation Experiment

To verify the impact of the mentioned methods on our model, the model without the channel attention module and sharpening function processing was used as the baseline model, comparing the baseline model with the two methods added separately. The experimental results are shown in Table 3 below, where baseline + CA method indicates that the CA module is added to the baseline model; baseline + sharpen (s1) and baseline + sharpen (s2) indicate that sharpening is added to only one student model in the baseline model, and baseline + sharpen indicates that sharpening is added to the baseline model for both student models.

The experimental results in Table 3 show that both the channel attention mechanism and the sharpening function play a role in improving the segmentation network. The results show that the semi-supervised loss in the experiments requires the pseudo-labeling of the teacher model to semi-supervise the two student models, and also requires the two student models to generate their own pseudo-labeling for cross-supervision. The single CA mechanism can lower the pseudo-label information entropy, and the sharpening function can improve the edge contour accuracy of the unsupervised prediction. Moreover, the combined use of the two methods can make the pseudo-labels on the teacher side and the student side more realistic, thus improving the semi-supervised learning efficiency and accuracy.

According to the experimental data in Table 1 and Table 2, we can see that when the proportion is 100%, most of the experimental results are better than the results compared to when the proportion of labeled images is 1/2. To ensure the segmentation accuracy on the premise of maximizing the reduction of the required label costs, we take the labeled image proportion from 10% to 100%, and increase the labeled images by 10% each time, and the impact of different labeled image proportions on the segmentation results is shown in Figure 6 below.

The dashed lines in Figure 5 represent the segmentation baselines of Unet; our model already outperforms the Unet network under supervised learning when the labeled image proportion is less than 50%, and the proportion has a greater effect on the results when the labeled image proportion is less than 50%. The result of Potsdam increases the most when the labeled image proportion is between 40% and 50%. The model improvement is most obvious for the Vaihingen dataset at 30% to 40% of the data, after which the model accuracy improves slowly as the labeled image proportion increases. The GID dataset shows a slight negative growth after the labeled image proportion exceeds 60%, which also proves that the over-computation of the method based on the entropy consistency constraint is not only cost-consuming but also leads to an increase in entropy. In conclusion, our model based on the dual cross-entropy consistency method achieves good segmentation results with 1/2 the labeled image proportion and significantly reduces the labeling costs.

## 4. Conclusions

We propose a semi-supervised remote sensing image semantic segmentation method based on dual entropy consistency to solve the problem of complex remote sensing image information and the large manual labeling cost required for remote sensing image segmentation tasks. Our teacher model incorporates a channel attention mechanism in the coding network of Unet to help the model to reduce the predictive information entropy of pseudo-labeling. Two student models share a coding network to ensure consistent input entropy, while sharpening the prediction results of the two student models to reduce the information entropy of unsupervised prediction and improve the accuracy of edge contours. The two student models need to be semi-supervised by the teacher model, as well as cross-supervising themselves. These two semi-supervised learning tasks based on entropy consistency alternately train the student models so that the student models can fully understand the information and minimize the entropy-increasing behavior in the prediction process. Simulation experiments show that the segmentation performance of our method on three publicly available remote sensing image datasets exceeds the segmentation accuracy of the current mainstream network models and reduces 50% of the labeled images, which indicates good generalizability. Subsequent work will optimize the model with respect to its computational complexity and training complexity.

## Figures and Tables

**Figure 1 entropy-25-00681-f001:**
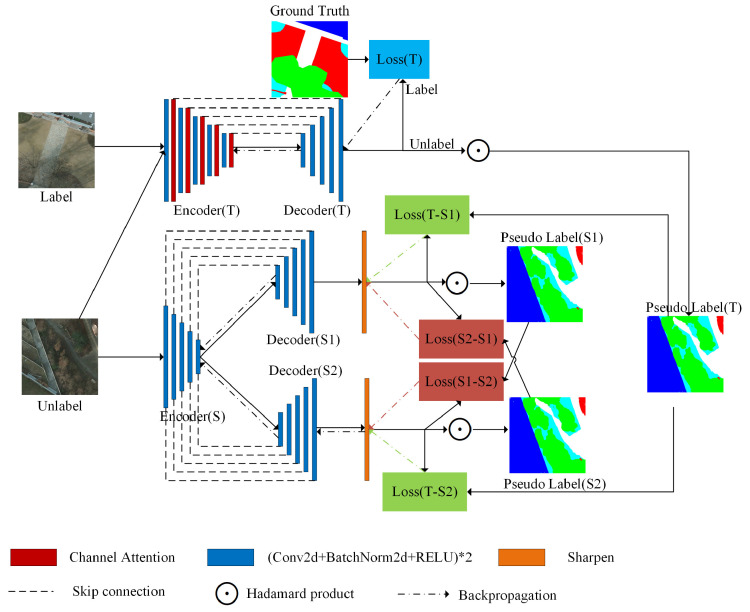
Semantic segmentation model based on dual consistent regularization.

**Figure 2 entropy-25-00681-f002:**
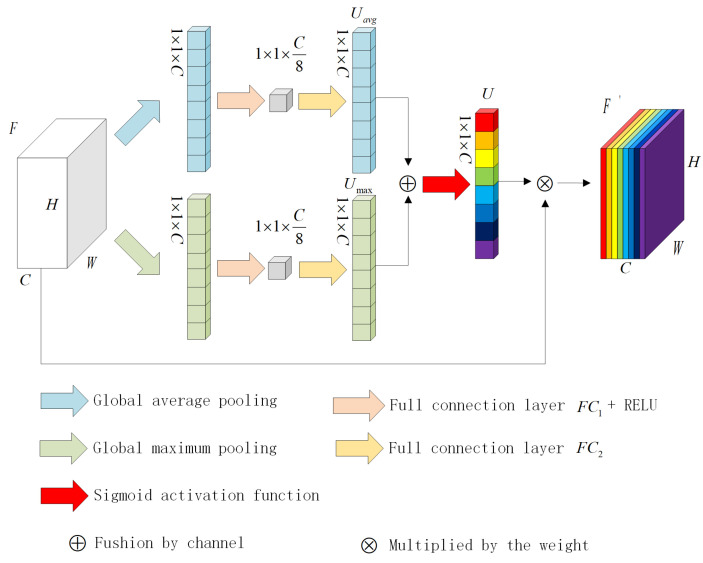
The CA mechanism.

**Figure 3 entropy-25-00681-f003:**
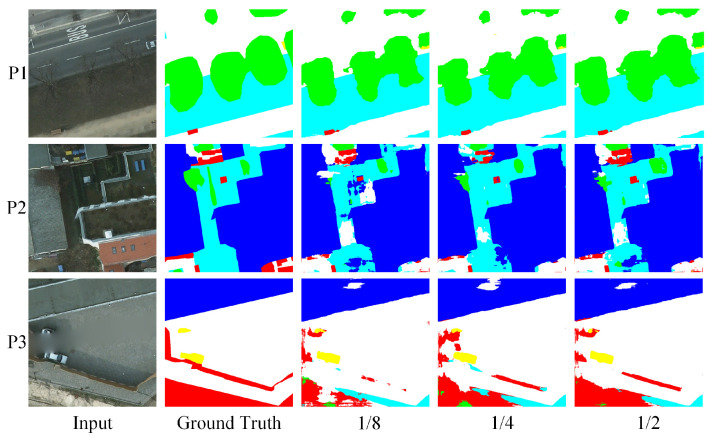
The predictions with different labeled image proportions on Potsdam.

**Figure 4 entropy-25-00681-f004:**
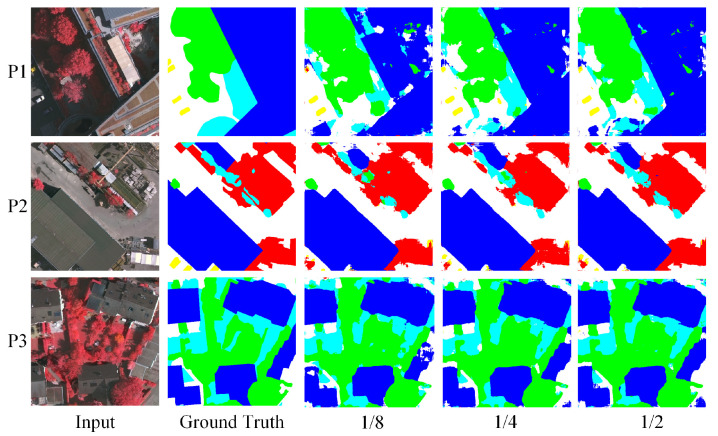
The predictions with different labeled image proportions on Vaihingen.

**Figure 5 entropy-25-00681-f005:**
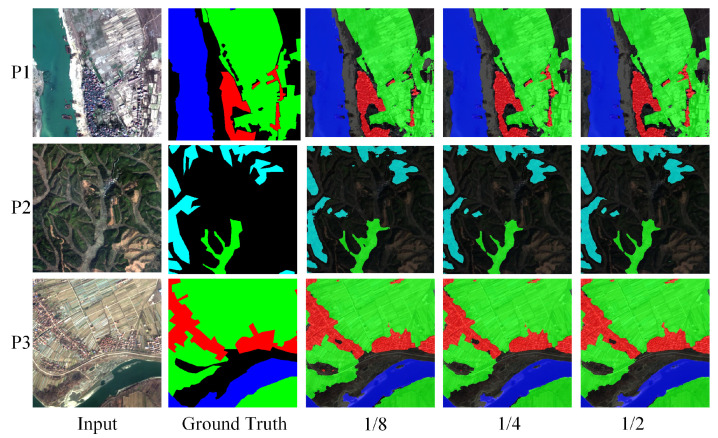
The predictions with different labeled image proportions on GID.

**Figure 6 entropy-25-00681-f006:**
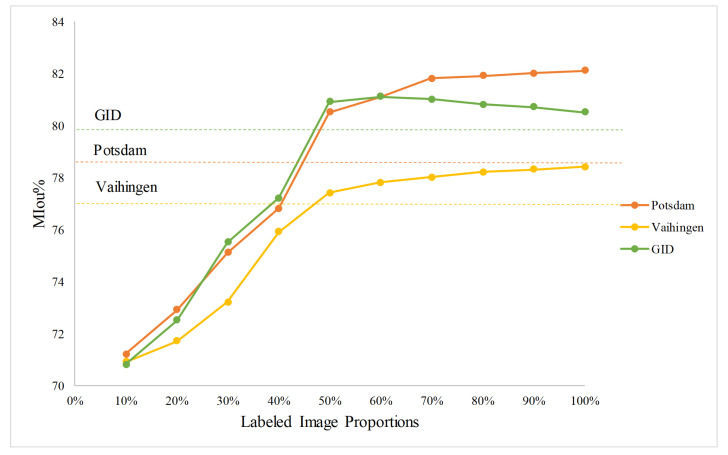
Ablation experiments of the effect of different labeled image proportions on the results.

**Table 1 entropy-25-00681-t001:** The MIou performance results for above methods on three datasets.

Dataset	Method	Labeled Image Proportion			
		1/8	1/4	1/2	1
Potsdam	Unet	-	-	-	78.2
	Attention-Unet	-	-	-	81.4
	U2-Net	-	-	-	81.3
	MeanTeacher	70.5	72.1	76.1	-
	CPS	**73.4**	75.2	77.8	-
	DST-CBC	73.3	**75.4**	78.3	-
	Our Algorithm	71.2	74.9	**80.5**	**82.1**
Vaihingen	Unet	-	-	-	76.8
	Attention-Unet	-	-	-	78.1
	U2-Net	-	-	-	78.3
	MeanTeacher	70.2	72.1	73.8	-
	CPS	71.7	72.8	74.0	-
	DST-CBC	**72.3**	73.4	74.9	-
	Our Algorithm	71.0	**73.5**	**77.4**	**78.4**
GID	Unet	-	-	-	79.8
	Attention-Unet	-	-	-	81.1
	U2-Net	-	-	-	81.2
	MeanTeacher	70.9	72.5	75.8	-
	CPS	**72.6**	75.7	76.4	-
	DST-CBC	72.4	75.1	76.5	-
	Our Algorithm	72.1	**76.3**	**81.8**	**82.1**

**Table 2 entropy-25-00681-t002:** The results for each category with different labeled image proportions on three datasets.

Dataset	Method	Category	Labeled Image Proportion			
			1/8	1/4	1/2	1
Potsdam	Our Algorithm	Impervious surface	75.2	77.1	83.6	**85.5**
		Building	76.9	81.8	86.3	**87.3**
		Low vegetation	67.1	71.8	76.4	**78.7**
		Tree	71.7	74.8	81.6	**82.3**
		Car	69.5	73.1	78.8	**80.7**
		Clutter	66.9	70.8	76.3	**77.9**
Vaihingen	Our Algorithm	Impervious surface	73.5	76.9	79.4	**81.9**
		Building	77.9	81.0	83.4	**85.2**
		Low vegetation	70.5	71.6	**76.2**	75.9
		Tree	72.1	73.9	**79.1**	78.3
		Car	59.2	62.2	66.5	**68.5**
		Clutter	72.8	75.4	79.8	**80.6**
GID	Our Algorithm	Farmland	73.7	77.6	82.5	**84.1**
		Forest	79.2	83.0	**88.2**	87.7
		Building	77.2	81.2	86.5	**86.6**
		Meadow	59.5	64.4	70.9	**71.5**
		Water	71.1	75.3	**80.9**	80.5

**Table 3 entropy-25-00681-t003:** Ablation experiments of each method.

Dataset	Labeled Image Proportion	Method	MIou (%)
Potsdam	1/2	Baseline	75.8
		Baseline + CA	76.3
		Baseline + sharpen (s1)	77.2
		Baseline + sharpen (s2)	77.4
		Baseline + sharpen	78.1
		Our Algorithm	**80.5**
Vaihingen	1/2	Baseline	72.3
		Baseline + CA	72.9
		Baseline + sharpen (s1)	73.5
		Baseline + sharpen (s2)	73.4
		Baseline + sharpen	74.6
		Our Algorithm	**77.4**
GID	1/2	Baseline	75.3
		Baseline + CA	76.5
		Baseline + sharpen (s1)	77.2
		Baseline + sharpen (s2)	77.0
		Baseline + sharpen	79.1
		Our Algorithm	**81.8**

## Data Availability

The remote sensing images utilized in this study are freely available at https://www.isprs.org/education/benchmarks/UrbanSemLab/Default.aspx, https://www.isprs.org/education/benchmarks/UrbanSemLab/Default.aspx, https://x-ytong.github.io/project/GID.html.

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
