# Peer review of "Semi-Supervised Semantic Segmentation of Remote Sensing Images Based on Dual Cross-Entropy Consistency"

_entropy, 2023, doi:10.3390/e25040681_

Round 1
Reviewer 1 Report
This work generally looks good. But some aspects need to be improved.
1. In fig. 2, what does the symbol "C" mean?
2. What does the pixel's label class actually refer to? Is that a 0-1 label?
3. E_t, D_t are not explained in Eqs. (7) and (8).
4. It is stated that "the pseudo-label Yt is obtained by making Hadamard products of the input image x and Pt". Why can doing so generate the pseudo-label? If a pixel is predicted as 1 while the real pixel in x is 0, their product will be 0. Thus the pseudo-label of the pixel has nothing to do with the Yt.
5. How to make sure the trustworthy level of the pseudo-label?
6. Some important up-to-date literature is missing such as "Song X, Aryal S, Ting K M, et al. Spectral–spatial anomaly detection of hyperspectral data based on improved isolation forest[J]. IEEE Transactions on Geoscience and Remote Sensing, 2021, 60: 1-16."
Reviewer 2 Report
The article entitled “Semi-supervised semantic segmentation of remote sensing images based on dual Cross-Entropy Consistency” is well-written and will be of interest to readers of Entropy from my point of view. In spite of these and before its publication, I consider that authors should make the following changes to improve the quality of the manuscript. The changes suggested are as follows:
R47: “Tarvainen et al. [10] proposed …” should be “Tarvainen and Valpola[10] proposed …”.
R220: “the number of training iterations was 100epoch”, here should be a space after 100.
The results in Figure3-5 are difficult to describe, they should be numbered.
Figure6: The result on Vaihingen is lower than the others datasets. Can you explain why?
